# Coffee and Microbiota: A Narrative Review

Federico Rosa [1,*], Benedetta Marigliano [1], Sergio Mannucci [1], Marcello Candelli [2], Gabriele Savioli [3,4], Giuseppe Merra [5], Maurizio Gabrielli [2], Antonio Gasbarrini [1,2], Francesco Franceschi [1,2] and Andrea Piccioni [2,*]

[1]  Facoltà di Medicina e Chirurgia, Università Cattolica del Sacro Cuore, 00168 Rome, Italy; benemarigliano@hotmail.com (B.M.); sergiomannucci1992@gmail.com (S.M.); antonio.gasbarrini@policlinicogemelli.i (A.G.); francesco.franceschi@policlinicogemelli.it (F.F.)

[2]  Department of Emergency Medicine, Fondazione Policlinico Universitario, Università Cattolica del Sacro Cuore, 00168 Roma, Italy; marcello.candelli@policlinicogemelli.it (M.C.); maurizio.gabrielli@policlinicogemelli.it (M.G.)

[3]  Emergency Department, IRCCS Fondazione Policlinico San Matteo, 27100 Pavia, Italy; gabrielesavioli@gmail.com

[4]  PhD School in Experimental Medicine, Department of Clinical-Surgical, Diagnostic and Pediatric Sciences, University of Pavia, 27100 Pavia, Italy

[5]  Section of Clinical Nutrition and Nutrigenomic, Department of Biomedicine and Prevention, University of Tor Vergata, 00133 Rome, Italy; giuseppe.merra@uniroma2.it

*  Correspondence: federosa1991@gmail.com (F.R.); andrea.piccioni@policlinicogemelli.it (A.P.)

**Abstract:** Coffee is one of the most widely consumed beverages in the world, which has important repercussions on the health of the individual, mainly because of certain compounds it contains. Coffee consumption exerts significant influences on the entire body, including the gastrointestinal tract, where a central role is played by the gut microbiota. Dysbiosis in the gut microbiota is implicated in the occurrence of numerous diseases, and knowledge of the microbiota has proven to be of fundamental importance for the development of new therapeutic strategies. In this narrative review, we thoroughly investigated the link between coffee consumption and its effects on the gut microbiota and the ensuing consequences on human health. We have selected the most significant articles published on this very interesting link, with the aim of elucidating the latest evidence about the relationship between coffee consumption, its repercussions on the composition of the gut microbiota, and human health. Based on the various studies carried out in both humans and animal models, it has emerged that coffee consumption is associated with changes in the gut microbiota, although further research is needed to understand more about this link and the repercussions for the whole organism.

**Keywords:** coffee; coffee consumption; microbiota; gut microbiota; microbiome

## 1. The Gut Microbiota

The human organism operates in concert with trillions of symbiotic microorganisms.

The host and its symbionts are called "holobionts," and their collective genome is known as the "hologenome" [1].

With the completion of the Human Genome Project, new horizons have opened in microbiome research for a better comprehension of host–microbe interactions in the four major colonization sites of the human body: The gastrointestinal (as headliner), genitourinary, cutaneous, and pulmonary tracts [2].

The plasticity of the holobiont is provided by changes that occur mainly in the human genome and gut microbiome. In the past, characterization of the gut microbiota was done by cultivation methods.

However, most organisms are refractory to cultivation, as many of the colonic bacteria are anaerobic and cannot be cultured under aerobic conditions. Only 30% of intestinal bacteria have been characterized by this method [3].

Metagenomics has been defined as the application of modern genomic techniques to the direct study of microbial communities in their natural environment, bypassing the need for isolation and lab cultivation [4].

Real-time polymerase chain reaction (rtPCR) is the gold standard for detecting known and unknown microbes without cultivation.

Among the different marker genes, 16S ribosomal RNA (16SrRNA) represents the main method to census the community. The 16S rRNA is a part of the small subunit of the 70S ribosome, and it represents the preferred molecule for bacterial identification because it is universally distributed and contains both conserved regions that are identical for all bacteria and nine interspersed regions of short hypervariability that are unique to individual bacteria [5].

In addition to 16SrRNA sequencing, phylogenetic characterization can be performed by shotgun metagenomics. This method comprises the sequencing of the collective genome of the microorganisms present in a sample, after DNA extraction and shearing into small fragments (next-generation sequencing (NGS)). One of the main outcome measures is species diversity, defined as the actual number of different species represented in a dataset, often expressed as species richness (the number of different species represented in an ecological community) and species evenness (the relative abundance with which each species is represented in the community).

Together, they constitute alpha diversity [6].

Gut microbial ecology is dynamic, the more abundant the biodiversity of an ecosystem, the better its ability to resist perturbations from the external environment [7].

In fact, the competitive interactions of increased microbial species promote the stability of the gut microbiome, partially accounting for differences in individual responses to the same diet or drugs [8,9].

A healthy gut microbiome can be defined as the normal individual microbiota that maintains and propagates wellness in the absence of disease [10].

Most gut microbes reside in the colon, where they are present in concentrations of $10^9$–$10^{12}$ CFU/mL and include >1000 different species [11].

The collective microbiome is 150 times larger than the human genome, indicating the enormous number of processes in which gut microbes are involved [12].

The gut harbours a complex bacterial community that consists almost entirely of seven major numerical bacterial phyla found in the adult human gut (more than 70% of all microbes in the body): *Bacteroidetes* (Gram-negative anaerobes), *Firmicutes* (Gram-positive), followed by *Actinobacteria*, *Proteobacteria*, *Fusobacteria*, *Verrucomicrobia*, and *Cyanobacteria* [13].

About 90% of bacterial species in adults belong to *Firmicutes* and *Bacteroidetes*. Most species in the phylum *Bacteroidetes* belong to the genera *Bacteroides* and *Prevotella*. Bacterial species belonging to the phylum *Firmicutes* include the genera *Clostridium*, *Eubacterium* and *Ruminococcus* [14].

Alongside the bacteria are members of the Archaea kingdom, predominantly *Methanobrevibacter* species, which produce methane in the gut, and *Eukarya* such as the yeast *Candida*, microbial parasites such as *Entamoeba*, and macroparasites such as *helminths* [15].

Finally, viruses and bacteriophages also play a significant role in maintaining a healthy and balanced gut, contingent on mutualistic interactions between different species and associated substrate availability [16].

Structurally, the microbiota is organized into mucosa-associated microbiota and luminal microbiota. The former contributes more to host cell protection and gut barrier function than the luminal, due to direct interaction with gut-associated lymphoid tissues [17].

The establishment and colonization of the gut microbiota is a complex process. The microbiota begins to develop as soon as a baby passes through the birth canal, with important variables, such as breastfeeding versus artificial feeding, caesarean versus natural delivery, as well as the choice and timing of feeding, and environmental factors, such as hygienic conditions, number of siblings, kindergartens and schools, animals in the home,

and rural versus urban lifestyle, being important determinants with long-term effects (including immunity) [18].

The structure of the human gut changes with aging, with a stabilization of the microbiota environment being achieved at around 3 years of age. After this period, the composition of the microbiota begins to differentiate and acquires similarity to that of the adult (50%) [19].

In the elderly, there is a higher proportion of pathogenic enterobacteria and a lower proportion of probiotic *Bifidobacteria* [20].

In centenarians, the lifespan decreases, due to changes in *Firmicutes* enrichment, increased proinflammatory responses (mediated by TNF alpha, IL-6, and IL-8) and decreased abundance of the anti-inflammatory *Faecalibacterium prausnitzii* [21].

Regardless of taxonomic classification, the healthy intestine comprises three enterotypes that are related to dietary habits: *Bacteroides*, *Prevotella*, and *Ruminococcus*, with considerable interindividual variability [22].

Enterotype 1 is characterized by a dominance of *Bacteroides*, which has saccharolytic and proteolytic activities and is involved in the synthesis of biotin, riboflavin, pantothenate, and ascorbate [23].

Enterotype 2 is *Prevotella* dominant, which acts as a mucin glycoprotein degrader, and is involved in thiamine and folate synthesis.

Enterotype 3 is characterized by *Ruminococcus* dominance, which has mucin-degrading activity and transports sugars from the membrane [24].

Regardless of the enterotype, some microbial members serve as the "core microbiota," while others act as a "flexible pool". The latter contributes to host adaptation and is generally acquired from ingested food, water, and various components of the environment [25].

The exchange of genetic material between the nucleus and flexible pool confers to the host the ability to adapt to an environment or food habit [26].

Depending on the combination of predominant species, an individual has a specific microbiome fingerprint [27].

Numerous high-quality data from the Human Microbiome Project (HMP) of the United States and the Metagenomics of the Human Intestinal Tract (MetaHIT) of Europe have now demonstrated the beneficial functions of normal intestinal flora on health, down to the genetic level [28].

These include protective (Peyer's plaques and IgA secretion), metabolic, and structural functions that comprise vitamin production, synthesis of catecholamines from protein catabolism, lipid regulation, and production of short-chain fatty acids (SCFAs) that not only regulate gene expression, but are the fuel for epithelial cells [29].

Specifically, acetate serves as an energy source for peripheral tissues, supporting lipogenesis and cholesterol synthesis. Propionate is metabolized mainly in the liver, and butyrate serves as an energy source for colonocytes, producing ketone bodies with carbon dioxide, and stimulates gut enteroendocrine cells for leptin production from adipocytes, including the production of glucagon-like peptide-1 (GLP-1) in gut cells [30].

Nutritional and lifestyle behaviours are thus crucial players contributing to aging and human diseases, including metabolic (such as, type II diabetes, liver disease, and cardiovascular disease), immunological (such as, inflammatory bowel disease and type I diabetes), and neurological (such as, autism and multiple sclerosis) diseases [31].

The relationship between dysbiosis and disease is bidirectional: the application of gut-modifying therapeutic strategies, including prebiotics (e.g., contained in coffee and other plant foods), probiotics, and faecal microbiota transplantation, can contribute to human–microbiome symbiosis by promoting better health [32].

## 2. Coffee: The "Longevity Beverage"

Coffee is one of the most popular beverages in the world; it is estimated that more than 2 billion cups are drunk every day [33].

The largest coffee consuming states in the world are Brazil and the United States [34].

Various bioactive compounds are contained in coffee, among which polyphenols, such as the alkaloids contained in caffeine, caffeic acid in roasted coffee beans, and most significantly, chlorogenic acids in green beans, stand out in importance [35].

Caffeine's mechanism of action operates on several levels in the body, specifically acting as an adenosine receptor antagonist on the nervous system [36].

The universally best-known action of caffeine is that it is a powerful stimulant, being able to increase the user's attention and ability to concentrate [37].

The safe threshold of caffeine appears to be 400 mg per day [38].

The link between coffee consumption and the onset of diseases, such as Parkinson's disease, diabetes mellitus type 2, nonalcoholic fatty liver disease (NAFLD), and liver cirrhosis, as well as coffee's effects on intestinal motility, have also been extensively studied.

A dose-dependent inverse relationship between tea or coffee (including decaffeinated) consumption and the risk of type 2 diabetes has been described [39].

The risk of developing nonalcoholic fatty liver disease (NAFLD) has been shown to be inversely associated with coffee consumption [40].

Coffee consumption is also associated with a lower risk of developing liver cirrhosis [41].

A relationship between coffee intake and a reduced risk of Parkinson's Disease onset is described, although the underlying mechanism remains unclear [42].

Caffeine is a smooth muscle stimulator, and according to some work, its consumption is therefore associated with a reduction in constipation [43].

The most interesting evidence, however, comes from the relationship of caffeine consumption with all-cause mortality.

Coffee consumption is associated with a reduction in mortality from all causes [38].

One reason may be that healthy people use caffeine more than those with disease.

Other studies have shown that caffeine consumption is associated with a reduction in all-cause mortality, regardless of coffee consumption [44].

One of the downsides of caffeine consumption is that it can lead to an addictive condition [45].

In fact, there is a real caffeine withdrawal syndrome, characterized by symptoms such as fatigue, irritability, headache, and difficulty concentrating [46].

## 3. Materials and Methods

This narrative review includes about 20 papers, published in English in the past 14 years, on the topic of the complex interaction between coffee consumption and its effects on the gut microbiota.

We searched PubMed® and Google scholar® with the following keywords, "Coffee" and "gut microbiota".

We selected the articles that we considered most suitable for this type of work, focusing mainly on the most recent studies, and favouring the latest evidence regarding this fascinating topic.

## 4. Coffee and Gut Microbiota

We saw that there are multiple effects of coffee consumption on the human body, so we wondered whether some of these effects were mediated by alterations in the gut microbiota.

We collected and selected the studies that we considered to be the most important in this regard.

We present to you the most important studies on this topic, starting with those conducted in animal models and ending with those conducted in humans.

### 4.1. Animal Models

Most of the studies, as we shall see, were conducted in animal models.

Starting with the oldest reports, we see that they are still quite recent, and that the various research groups have focused on one of the different diseases for which coffee

consumption appears to be protective, to better understand the link between consumption of this beverage and the onset of disease.

Below, we will discuss the most important studies conducted on the topic, listed by publication date.

Starting from the observation that coffee consumption is negatively correlated with the onset of type 2 diabetes, researchers investigated the link between a high-fat diet and coffee consumption in rats, and found that coffee consumption succeeded in changing the gut microbiota of rats fed the high-fat diet [47].

In another work conducted in mouse models, researchers witnessed a reduction in nonalcoholic fatty liver disease (NAFLD) in caffeine-consuming mice, accompanied by changes in the gut microbiota [40].

In a third study, mice were given caffeic acid, one of the main phenolic acids found in coffee [36].

Following caffeic acid supplementation, changes were found in the microbiota, such as an increase in *Akkermansia* and *Dubosiella*, and in re-examining the relative abundance, an increase in *Alistipes* and a decrease in *Turicibacter* and *Bacteroides* was documented [48].

In this interesting study conducted in rodents, researchers questioned the relationship between coffee consumption and glucose metabolism, studying the effects of chlorogenic acid (CGA), a polyphenol contained in coffee, on the microbiota [36].

The results were surprising. Chlorogenic acid (CGA) led to a change in the microbiota accompanied by an increase in short-chain fatty acid (SCFA) producers with a protective role towards the intestinal barrier [49].

In a recent study in PSD (paradoxical sleep deprivation) rats, there was a change in the gut microbiota after the administration of coffee and decaffeinated coffee [50].

In another interesting study in mouse models, researchers focused on the effect of coffee consumption on aspirin absorption.

Coffee bean extract (CBE) was administered to rodents, which resulted in a change in their gut microbiota such that there was an increase in the populations of *Lactobacillaceae* and *Muribaculaceae* and a decrease in *Bacteroidaceae*, *Proteobacteria*, and *Helicobacteraceae* [51].

Staying with animal models, we mention this recent and very interesting study in mice, in which the effect of coffee consumption and subsequent sleep restriction on the composition of their gut microbiota was investigated.

No changes were found in *Bacteroidetes* and *Firmicutes*, but, on the contrary, *Actinobacteria* and *Proteobacteria* were decreased.

So, in this work, caffeine administration resulted in a change in the gut microbiota of mice [52].

### 4.2. Studies on Humans

Having told you about these important studies conducted in animal models, it is now time to get to those conducted in human models.

Let us start with this study that is somewhat older than the others, with a very small number of participants (16) whose faecal samples were collected before and after a moderate intake of coffee (three cups a day for three weeks). Coffee consumption was found to be associated with an increase in *Bifidobacterium* spp. without affecting the dominant microbiota, however, accompanied by an increase in metabolic activity [53].

We then mention this interesting study in which chlorogenic acids, the most important bioactive compounds in coffee, were administered in addition to caffeine [36].

The assumptions of this study, as already explained in the introductory part of our article, are that coffee consumption is inversely related to the occurrence of both type 2 diabetes mellitus and nonalcoholic fatty liver disease (NAFLD) [39,40].

In this study, researchers showed that administering caffeine plus chlorogenic acids to a group of patients with diabetes and nonalcoholic fatty liver disease resulted in a reduction in their weight, probably related to an increase in intestinal bifidobacteria [54].

In this work, patients were divided according to their degree of coffee consumption, and it was found that higher levels of *Prevotella*, *Porphyromonas*, and *Bacteroides* were found in heavy coffee drinkers [55].

In a more recent study conducted on a small number (30) of healthy volunteers, it was found that coffee administration led to alterations in the gut microbiota, although this was not significant [56].

In this latest work, scholars focused on chlorogenic acids, the main polyphenols contained in coffee, and it was found that they altered the composition of the microbiota in the healthy volunteers in this important and interesting study [57].

An additional supportive action of coffee could be changes in the composition and metabolic function of the gut microbiota by polyphenols and other undigested prebiotic constituents of coffee (e.g., polysaccharides and melanoidins) [58]. Observational data assessed how dietary fibre is rapidly metabolized into short-chain fatty acids (SCFAs), resulting in up to a 60% increase in the *Bacteroides/Prevotella* bacterial group after consumption of coffee prepared from medium roasted Arabica beans [59].

In another experiment conducted in mouse models, the modulating action of coffee toward the gut microbiota was confirmed; in fact, there was a decrease in *Clostridium* spp. and *Escherichia coli* and an increase in *Bifidobacterium* spp. [60].

Selective metabolism and amplification of some bacterial populations following coffee consumption appear to be mainly due to its richness in polyphenols [61].

In spite of these promising results, much more clinical research is needed to clarify the impact of long-term coffee intake on gut microbiota composition and its health implications.

As for our narrative review, we saw how the interest of researchers has been steadily increasing in recent years, and how, although most of the studies have been conducted in animal models, there is an increase in the number of trials that have humans as protagonists.

We have also collected all of the most important studies and considerations in this summary table (Table 1).

**Table 1.** The most important studies and considerations on the relationship between coffee consumption and gut microbiota.

| Studies Conducted on Animals | | |
|---|---|---|
| **Author** | **Comments** | **Year** |
| **Nakayama et al. [60]** | **In mice that consumed coffee and galacto-oligosaccharides, an increase in *Bifidobacterium* spp., a decrease in *E. coli* and *Clostridium* spp.** | **2013** |
| Cowan et al. [47] | Coffee consumption succeeded in changing the gut microbiota of mice fed a high-fat diet | 2014 |
| Vitaglione et al. [40] | Evidence of reduced nonalcoholic fatty liver disease (NAFLD) in caffeine-consuming mice, accompanied by changes in gut microbiota. | 2019 |
| Wan et al. [48] | Administration of caffeic acid supplement in rats resulted in an increase in *Akkermansia* and *Dubosiella*, an increase in the abundance of *Alistipes*, and a decrease in the abundance of *Turicibacter* and *Bacteroides*. | 2021 |
| Ye et al. [49] | Chlorogenic acid (CGA) administration in mice led to a change in the microbiota accompanied by an increase in short-chain fatty acid (SCFA) producers with a protective role toward the intestinal barrier. | 2021 |
| Gu X et al. [50] | Even the consumption of decaffeinated coffee was found to affect the composition of the gut microbiota in PSD (paradoxical sleep deprivation) rats. | 2022 |
| Kim et al. [51] | After coffee bean extract (CBE) intake in mice, gut microbiota showed an increase in the populations of *Lactobacillaceae* and *Muribaculaceae* and a decrease in *Bacteroidaceae*, *Proteobacteria* and *Helicobacteraceae*. | 2022 |
| Song et al. [52] | In mice with chronic caffeine-induced sleep restriction, *Actinobacteria* and *Proteobacteria* decreased. | 2022 |

**Table 1.** *Cont.*

| | Studies Conducted in Humans | |
|---|---|---|
| **Author** | **Comments** | **Year** |
| Jaquet et al. [53] | Administration of 3 cups of coffee per day for 3 weeks to 16 healthy adult volunteers, resulted in an increase in *Bifidobacterium* spp. but did not affect the dominant microbiota. | 2009 |
| Ludwig et al. [57] | Chlorogenic acids, the main polyphenols contained in coffee, were found to change the composition of the microbiota in 3 healthy volunteers. | 2013 |
| Mansour et al. [54] | Administration of caffeine plus chlorogenic acid to a group of 26 patients with diabetes and nonalcoholic fatty liver disease, resulted in a reduction in their weight, probably related to an increase in intestinal *Bifidobacteria*. | 2020 |
| González et al. [55] | Higher levels of *Prevotella, Porphyromonas,* and *Bacteroides* were found in heavy coffee drinkers in a study of a group of 147 individuals. | 2020 |
| Chong et al. [56] | In a group of 30 healthy participants, coffee administration led to alterations in the gut microbiota that did not attain significance. | 2020 |

## 5. Discussion

We decided to address this fascinating topic for several reasons.

Coffee is one of the most widely drunk beverages in the world [62], and consequently thoroughly understanding its effects on humans is one of the topics that fascinates multiple researchers.

As we have seen, coffee consumption appears to be implicated in a wide variety of diseases, and its protective role against Parkinson's disease, type 2 diabetes mellitus, non-alcoholic fatty liver disease (NAFLD), and liver cirrhosis has been described [39–42].

To be honest, however, what is most striking is how coffee consumption is associated with a reduction in all-cause mortality [38], even though, as explained earlier, the most-accepted current theory is that healthy people are greater consumers of this beverage than unhealthy people.

Since it is difficult to investigate this important and fascinating relationship, scholars have therefore focused on the link between coffee consumption and the occurrence of specific diseases.

However, since coffee is a food, according to many authors, there could be a close link with its repercussions on the gut microbiota.

The gut microbiota is one of the most evolving fields of medicine in recent years [63], which is increasingly fascinating researchers worldwide.

There are many reasons for this, starting with the increased technological development that has made it less complicated to conduct studies on this topic.

It is now well known how lifestyle (e.g., smoking) affects the composition of the gut microbiota [64], and again one of the reasons we have devoted ourselves to this topic is precisely to thoroughly investigate this habit, which is as widespread in the world as coffee drinking.

Knowing the gut microbiota may also prove to be of fundamental importance as a potential new therapeutic strategy, as is the case with *Clostridium difficile* infection, for example [65].

Scholars are increasingly focusing on the link between gut microbiota and diseases even far removed from the gastrointestinal system [66].

For this reason, we focused on the effect of coffee consumption on the composition of the gut microbiota, precisely because, in the future, we might try to exploit this as a starting point for new potential therapies.

In most of the literature, researchers started from an already known hypothesis or link, such as the protective effect of coffee consumption against various diseases, such as Parkinson's, type 2 diabetes mellitus, nonalcoholic fatty liver disease (NAFLD), and liver

cirrhosis [39–42], proceeding then to come to an understanding of its mechanism of action, or to determine whether there was an underlying link or change in the gut microbiota.

We see how the studies we have collected are all recently published, with the oldest one having been published in 2009.

Most were conducted on animal models, probably due to the fact that they are easier to perform than those conducted on a human population.

There is no doubt that coffee consumption leads to changes in the gut microbiota, but there is no unanimity regarding the composition of individual species of bacteria.

In all of the studies we selected, both in those conducted in animal models and in those conducted in humans (Table 1), it was found that in some way coffee consumption affects the composition of the gut microbiota. In some of them, a change was found in a specific microbial species, while in others only generic alterations to the gut microbiota were found.

In several studies, there was a decrease in *Proteobacteria* [51,52], and an increase in *Akkermansia* [48].

We then mention this narrative review, in which the role of several nutrients, including polyphenols, contained in coffee is analysed [67].

Polyphenols have demonstrated the ability to modulate the gut microbiota by increasing the concentration of *Faecalibacterium* sp., *Lactobacillus, Akkermansia,* and *Bifidobacterium* associated with short-chain fatty acid (SCFA) production [67].

Going into the specifics of the selected papers (Table 1), we see that most of them (8) were conducted on animal models, in this case always murine, while only 5 were conducted on humans.

As for those conducted on animals, we see that they are all interventional, although there is important heterogeneity in the compounds administered and on the type of population in which it took place.

Different compounds were administered in the various studies, such as chlorogenic acid [49], caffeic acid supplement [48], coffee bean extract (CBE) [51], and caffeine [40,52], while in other studies, coffee, understood in a generic sense, including decaffeinated [47,50] or coffee and galacto-oligosaccharides [60], was used.

The murine population involved included both common individuals [48,49,51,60] fed a high-fat diet [47], PSD (paradoxical sleep deprivation) rats [50], and mice with chronic caffeine-induced sleep restriction [52].

In light of this heterogeneity, regarding both the compounds administered and the type of population studied, the one common element that emerged is how coffee consumption alters and has effects on the gut microbiota.

Because it is such a complex topic with so many variables, many more studies on the subject are certainly needed, but we believe that this can be an important starting point.

As for the studies conducted in humans, only one was observational, while the others were interventional.

The observational study was the only one that involved a substantial number of people (147), and concluded that heavy coffee drinkers had a different microbial composition [55].

Regarding the other interventional studies, we see how the heterogeneity of the substances administered is present, as coffee was administered in some [53,56] and chlorogenic acid in others [54,57].

Healthy people were involved in most human studies [53,56,57], while one study focussed on patients with diabetes and nonalcoholic fatty liver disease [56].

All of the interventional studies had the major limitation of having a small number of people involved, ranging from a minimum of three [57] to a maximum of 30 [56].

These studies are certainly much more significant because they were conducted in humans rather than animal models, and despite the various limitations exhibited, such as the low number of participants, the different type of substances administered, and variation in the health status of participants, it was unanimously shown that coffee consumption alters the composition of the gut microbiota.

Whether this alteration is what determines health effects on the individual is not yet certain, but the current evidence does seem to suggest it.

What unites both animal model and human studies is the fact that it is a very broad field, since a large number of compounds are collected under the generic term of coffee, and also the population sample studied has not always been represented by healthy individuals. Despite this it is obvious that coffee consumption has effects and modifies the gut microbiota.

Therefore, we expect that in the future more studies will be conducted on this topic, diversified according to the type of substance or compound examined, and according to the type of population under study, so as to investigate step by step, this relationship that, at present, appears to undoubtedly link coffee consumption to the gut microbiota.

This work has several limitations, the most important being that it is a review and therefore not supported by experimental data, and for several topics, there are not enough studies available to state a hypothesis accurately.

Unfortunately, at the present time, there are no mechanistic or otherwise detailed studies that provide in-depth investigation of the link between coffee consumption and the gut microbiota, and this is certainly one of the major limitations of this work and of the research conducted on this topic, but we hope that these data may also be present in the near future. Most of the studies were conducted by analysing the gut microbiota before and after coffee administration, or by comparing the gut microbiota of coffee drinkers with a control group.

Another limitation we feel compelled to make is that the term coffee may be too general, since it encompasses a vast number of compounds within it, such as theophylline, which we will discuss later. The purpose of our narrative review is to investigate more of the fascinating world enveloping the gut microbiota.

For this reason, we decided to focus on one of the topics that we believe to be of greatest interest, in which, however, the various studies were conducted by administering or observing the effects of coffee understood in a generic term and not subdivided according to its metabolites, which is why we also had to maintain this approach, with the hope that in the future more specific and diverse studies will be conducted for the various compounds encompassed under the generic term of coffee.

Theophylline is one of the most discussed purine metabolites contained in coffee, but not exclusively, being present in large concentrations in cocoa as well [68]. For this reason, among the limitations of this research we must unfortunately include the fact, that at present, it is not possible to discern the extent to which the considerations made here are valid only for individual metabolites, since the various studies were conducted on the administration or intake of coffee and not on individual metabolites. We believe that theophylline is certainly one of the most important compounds present in coffee. Specific studies on its interaction with the gut microbiota would be of great interest.

The major strength of this paper is that it is up to date with the latest evidence regarding this very complex and fascinating topic.

We have carefully selected the articles that we considered most suitable for this type of coverage, focusing on the topics that are currently attracting the most interest from researchers.

## 6. Conclusions

In conclusion, we believe that the topic we have just discussed is as fascinating as ever, since it brings together two of the most interesting topics in modern medicine.

On the one hand we have coffee consumption, which we have seen many times as being one of the most widespread habits in the world [69], and therefore, involving a very large number of people, with important repercussions in the health of the individual, not just to the gastrointestinal system.

On the other hand, we have the gut microbiota, one of the areas in which there is an increasing interest from researchers all over the world, an ever-expanding field, constantly searching for new therapeutic weapons against various diseases that afflict humans [65].

In this review, we have selected and collected the most important studies conducted on this topic, dividing them between those conducted on animal models and those on humans, going deep into the relationship between coffee consumption and the repercussions on the gut microbiota, and their consequences on human health.

This research has shown how we have only begun to delve into this topic in the last fourteen years, with too few studies to be able to provide solid evidence in this regard.

What has emerged from our research is that coffee consumption exerts numerous effects on different systems, and that it acts by modifying the composition of the human gut microbiota, although at the moment not all studies have come to the same conclusions, so the hypothesis that these effects on different organs are mediated by changes in the gut microbiota remains valid.

Our hope is that we will be able to investigate this relationship more thoroughly, so that in the future it may become a new therapeutic strategy for the treatment of different diseases, going directly to the gut microbiota, as has been happening, for example, in the case of *Clostridium difficile* infection [65].

Therefore, this work represents a valuable starting point for conducting new and increasingly important studies directed at fully dissecting this very fascinating topic.

**Funding:** This research received no external funding.

**Institutional Review Board Statement:** The study did not require ethical approval.

**Informed Consent Statement:** Not applicable.

**Data Availability Statement:** Data available on request due to restrictions, the data presented in this study are available on request from the corresponding author.

**Conflicts of Interest:** The authors declare no conflict of interest.

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
