# Peer review of "Coffee and Microbiota: A Narrative Review"

_cimb, doi:10.3390/cimb46010057_

Round 1

Reviewer 1 Report

Comments and Suggestions for Authors

This Review is interesting as a topic, but it is written and described schematically and lacks links and discussion, such as in section 2. COFFEE: THE "DRINK OF LONGEVITY

 The data are provided by the documents that the authors have collected for reference, without these experimental data being linked and commented on by the authors of the manuscript. In addition, even the term coffee is too general for a scientific article.

 Although the authors have specified that the manuscript is "a narrative review," the manuscript does not appear balanced, and the data reported are not commented on.  In addition, any limitations of the experimental model chosen and presented are not always indicated.

This minireview does not propose any comments or possible applicability.  It has not been disclosed to whom this review might be of interest and who might benefit from reading the manuscript.

Section 7. Section 7. DISCUSSION should be more annotated and analyzed so that the role of the authors is evident to the reader.  

The manuscript type is a review and should go beyond simply listing the selected papers.

 Abbreviations and acronyms should be checked and spelled out in full.

 This manuscript covers a topic of possible interest, but results in a list of selected articles. 

The authors' expertise, if any, on the microbiome and coffee compounds do not appear in the manuscript.

  I ask Authors to implement the manuscript with critical comments, i.e., editorial review to make the manuscript valuable to readers

Comments on the Quality of English Language

English language is fine

Author Response

We thank you for your valuable comments, we hope we have greatly improved our manuscript, we are available to clarify any doubts and further improve the article.

We thank you for your valuable suggestions.

We added the fact that the term coffee is too general for a scientific article among the limitations of this study, but since it is also used in the other studies conducted on the topic, we preferred to keep it.

We have added a large section in the discussion chapter in which we carefully comment on the various selected articles, describe their limitations and the conclusions reached.

We checked and spelled out abbreviations and acronyms in full.

Regarding our expertise on the microbiome, we are a team of researchers many activities on the topic, just look at our publications or the fact that we were chosen as guest editors for special issue on the gut microbiota https://www.mdpi.com/journal/biomedicines/special_issues/N5VIYC0888, but we do not think it is fair to include our expertise in the article, instead leaving space only for what there is to be expounded on the topic.

We thank you for the valuable suggestion, we have implemented the manuscript with critical comments, that is, an editorial review to make the manuscript valuable to readers, as you suggested. We remain available for any comments and to refine and make this article even better.

Reviewer 2 Report

Comments and Suggestions for Authors

This interesting review examines the effects of coffee consumption on the intestinal microbiome, which is a rarely examined feature. I have some remarks:

- Table 1: The human studies should mention whether they were observational or interventional.

- Paragraph 3 and 4 could be summarized in a scheme instead.

- Discussion: You looked at caffeine in coffee, but the theofyllin should also be discussed.

Comments on the Quality of English Language

Some small grammatical errors and sense construction to be corrected throughout the text.

Author Response

We thank you for your valuable comments, we hope we have greatly improved our manuscript, we are available to clarify any doubts and further improve the article.

This interesting review examines the effects of coffee consumption on the intestinal microbiome, which is a rarely examined feature. I have some remarks:

- Table 1: The human studies should mention whether they were observational or interventional.

We added it, we specified how the only observational one was the one conducted by Gonzalez et al.

- Paragraph 3 and 4 could be summarized in a scheme instead.

Thanks for the suggestion, we have rearranged the paragraph breakdown as suggested.

- Discussion: You looked at caffeine in coffee, but the theofyllin should also be discussed.

Thank you for the excellent point, theophylline being only one of the many compounds present in coffee, although it is one of the most important ones, it is not possible at present to specify what its single effect is on the gut microbiota, so we have added this extended reflection in the limitations of our study, thanks again for the cue.

Reviewer 3 Report

Comments and Suggestions for Authors

1)    Manuscript language needs further improvement.

2)    Abstract is not impressive and need improvement.

3)    Conclusion need to be very clear and should reflect the title of the manuscript.

4)    The author mentioned regarding coffee consumption but didn’t mention which animal models used.

5)    The author must include clear link between coffee consumption and gut micro- biota.

6)    The author must include some mechanistic study which can show the link between coffee consumption and gut micro biota.

Comments on the Quality of English Language

1)    Manuscript language needs further improvement.

2)    Abstract is not impressive and need improvement.

3)    Conclusion need to be very clear and should reflect the title of the manuscript.

4)    The author mentioned regarding coffee consumption but didn’t mention which animal models used.

5)    The author must include clear link between coffee consumption and gut micro- biota.

6)    The author must include some mechanistic study which can show the link between coffee consumption and gut micro biota.

Author Response

We thank you for your valuable comments, we hope we have greatly improved our manuscript, we are available to clarify any doubts and further improve the article.

1)    Manuscript language needs further improvement.

We did it

2)    Abstract is not impressive and need improvement.

We modified it as requested

3)    Conclusion need to be very clear and should reflect the title of the manuscript.

Thanks for the suggestion, we have executed it

4)    The author mentioned regarding coffee consumption but didn’t mention which animal models used.

We added them.

5)    The author must include clear link between coffee consumption and gut micro- biota.

 We added it. In all the studies we selected, both in those conducted in animal models and in those conducted in humans, (Table 1) it was found that in some way coffee consumption affects the composition of the gut microbiota, in some of them a change was found to a specific microbial species, while in others only generic alterations to the gut microbiota.

6)    The author must include some mechanistic study which can show the link between coffee consumption and gut micro biota.

Unfortunately, we could not find a mechanistic study that could prove the link between coffee consumption and the gut microbiota, and we also added it in the text among the limitations, also discussing the mechanism of the other types of studies found.

Round 2

Reviewer 1 Report

Comments and Suggestions for Authors

The authors addressed most of the required comments and changed the manuscript accordingly. The manuscript sounds now better.

Comments on the Quality of English Language

english language is good